# Emission factors for Vietnamese beef cattle manure sun-drying and the effects of drying on manure microbial community

Van Thanh Nguyen[1], Koki Maeda[2]*, Yukiko Nishimura[2], Trinh Thi Hong Nguyen[1], Kinh Van La[1], Dien Duc Nguyen[3], Tomoyuki Suzuki[2,4]

1 Institute of Animal Sciences for Southern Vietnam, Di An, Binh Duong, Vietnam, 2 Crop, Livestock & Environment Division, JIRCAS, Tsukuba, Ibaraki, Japan, 3 Faculty of Animal Science -Veterinary Medicine, Tay Nguyen University, Buôn Ma Thuột, ắk Lắk, Vietnam, 4 Institute of Livestock and Grassland Science, NARO, Nasu-shiobara, Tochigi, Japan

* k_maeda@affrc.go.jp

## Abstract

Livestock manure and its management are significant sources of greenhouse gas (GHG). In most Southeast Asian countries, the current GHG emissions are estimated by the Intergovernmental Panel on Climate Change (IPCC) Tier 1 approach using default emission factors. Sun-drying is the dominant manure treatment in Vietnam, and in this study, we measured GHG emissions during manure drying using a chamber-based approach. Results show the emission factors for $CH_4$ and $N_2O$ were $0.295 \pm 0.078$ g $kg^{-1}$ volatile solids (VS) and $0.132 \pm 0.136$ g $N_2O$-N $kg^{-1}$ $N_{initial}$, respectively. We monitored the total bacterial/archaeal community using 16S rRNA gene amplicon sequencing and measured the abundance of functional genes required for methanogenesis (*mcrA*), nitrification (*amoA*) and denitrification (*nirK*, *nirS* and *nosZ*) processes. Methane emission occurred only at the beginning of the drying process (days 1 to 3). The results of amplicon sequencing indicated that the relative abundance of methanogens also decreased during this period. Although some nitrification activity was detected, there was no significant $N_2O$ emission. These findings well describe the manure management system in south Vietnam and the GHG emission from this manure category, paving the way for higher Tier estimations using country-specific values.

## Introduction

Livestock production is increasing rapidly (especially in developing countries including those of Southeast Asia) due to economic growth with higher personal incomes [1]. Among the Southeast Asian countries, Vietnam ranks third among the Association of South-East Asian Nations (ASEAN) countries; Vietnam had 5.8 million head of cattle in 2018 [2], >90% of which were beef cattle, and its livestock production contributed approx. 32% of the national agriculture sector gross domestic product (GDP) in that year [3]. With the growth of the livestock industry in Vietnam, roughly 20 million tons of manure are now excreted by cattle each year. This amount of livestock manure has led to significant environmental concerns, such as

**Data Availability Statement:** All relevant data are within the paper and its Supporting Information files.

**Funding:** The author(s) received no specific funding for this work.

**Competing interests:** The authors have declared that no competing interests exist.

eutrophication and groundwater pollution [4]. One of the major concerns is the emission of greenhouse gases (GHG) such as methane ($CH_4$) and nitrous oxide ($N_2O$) [5]. The total GHG emission from ruminant livestock was estimated to be 2.72 Gt $CO_2$-eq, which accounted for 47%–54% of all non-$CO_2$ GHG emissions from the global agricultural sector [6] and approx. 9% of total GHG emissions globally [7].

Both $CH_4$ and $N_2O$ can be produced by the activity of microbes in manure. Methane can be produced by methanogenic archaea under strictly reducing conditions, with three main substrates: $CO_2$, acetate, and other methyl groups [8]. One major methanogenic pathway is hydrogenotrophic methanogenesis ($CO_2$ reduction with $H_2$ as an electron donor), which is the dominant pathway for the process of fiber digestion in the cattle rumen [9]. Another pathway, aceticlastic methanogenesis, is sometimes dominant in man-made ecosystems such as manure digesters [10].

Nitrous oxide can be produced through a nitrification-denitrification process that can occur on the abundant inorganic nitrogen in manure. Nitrification occurs aerobically; it converts ammonia into hydroxyl amine via ammonia monooxygenase, hydroxyl amine into nitrite via hydroxylamine oxidoreductase and nitrite into nitrate via nitrite oxidoreductase [11]. Nitrifiers produce $N_2O$ during the oxidation of hydroxylamine to nitrite. Denitrification occurs anaerobically with a stepwise reduction from nitrate into nitrite, nitric oxide, nitrous oxide, and dinitrogen [12]. Functional genes that are required for methanogenesis, nitrification, and each step of denitrification can be used as molecular markers to help understand the ecology of the microbes that are responsible for GHG emission in environmental samples [13–17].

In Vietnam, the majority of beef cattle manure is collected and spread on land for sun-drying for 3–4 days [18, 19]. The GHG emissions from Vietnamese livestock manure management, including during sun-drying, are estimated by the Intergovernmental Panel on Climate Change (IPCC) Tier 1 approach. A default emission factor (EF) provided by the IPCC (1.5%–2.0% for methane and 2% for $N_2O$) is used for this approach [20], but the use of the default EF could lead to an incorrect estimation of the nationwide GHG emission in Vietnam, because many factors, including climate, cattle breed and feedstuffs, differ significantly among countries and across regions. Thus, there is a need for a country specific national emission factor to estimate nationwide GHG emissions from manure. In this regard, individual countries, and especially developing countries, have been encouraged to develop an EF that reflects their specific conditions, as doing so enables each country using the Tier 1 approach to step up to a Tier 2 approach [21].

We conducted the present study in order to: (1) identify the major manure management method(s) in Vietnam by distributing a farm survey, (2) measuring the GHG emission from the beef cattle manure sun-drying process by a chamber-based method, in order to provide the country-specific data for this category as required for a Tier 2 approach, and (3) monitored the changes in the microbial community and the abundance of functional genes that are required for microbial GHG production over time in order to understand the pattern of GHG emission.

## Materials and methods

### Survey of the beef cattle farmers in Bentre province

We conducted a farm survey to identify the major manure management methods in Vietnam. Twenty typical beef cattle farms in Vietnam's Bentre province were chosen as representative farms. The survey gathered information about each farm's size, labor resources, number of cattle, cattle breed and body weight, land resources for feed production, amount and composition

of feed, and type(s) of manure management. The individual pictured in S2 Fig has provided written informed consent (as outlined in PLOS consent form) to publish their image alongside the manuscript.

## Manure sun-drying experiments and GHG emission measurement

We conducted 7-day manure sun-drying experiments at the Institute of Agricultural Science for Southern Vietnam (IASVN) experimental station (Binh Duong, 11° 13' 36.2" N, 106° 36' 54.9" E), following the producers' practice as identified by the survey. These experiments were approved by the "Science and Technology Committee of IASVN". To reproduce the producers' practice, we used 76.5 ± 5.0 kg of manure for Run 1 and 100 ± 0 kg for Run 2, with two replicates. Fresh manure was collected at the IASVN station immediately following excretion; urine was separated by the angle floor drain in the pen and eliminated from the manure used for the experiment. The manure from the cattle barn was derived from a beef cattle herd of 58 Brahman cattle and 26 calves fed the same diet, which included guinea grass, concentrate, and cassava waste. The chemical composition of the manure collected is summarized in S4 Table.

The manure was mixed well and spread on plastic sheets which were placed in the yard of the cattle barn so that the manure density was 14 kg fresh manure/m$^2$. We had two replicates (chambers) for each run, and the same experiments were done twice (Runs 1 and 2). During the 7 days experimental period, the manure was dried in the sun all day and covered all night with tarpaulins. No rain fell during the experimental period. Manure samples were collected at 12:50 p.m. every day at five points (four at corners and one at center) in each chamber, mixed well, and kept in a freezer at −20°C for until analysis. The weights of the samples were recorded every time to enable the calculation of the loss by sampling precisely. The weights of the total manure on day 0 and day 7 and those of the samples were recorded to calculate the reduction in manure weight.

A polyvinyl chloride (PVC) chamber (3 m long, 3 m wide, 2 m high) equipped with an air-blowing ventilator and a gas sampling port on the ventilation exhaust was used for the GHG emission measurements (S1 Fig), as described previously [22]. The airflow was kept constant throughout the experimental period by an inverter, and fresh air was introduced under the skirt of the chamber. Manure was covered by the chamber twice daily from 6:00 to 8:00 and from 13:00 to 15:00 for the gas sampling. Gas samples were taken from the sampling port using a 20 ml syringe and put into pre-vacuumed 10-ml vials at 8:00 and 15:00 with two replicates for seven days. Ambient air samples were also collected for the background measurement.

The $CH_4$ and $N_2O$ concentrations in the vials were measured by a gas chromatography (GC) device equipped with a flame ionization detector (FID) and an electron capture detector (ECD) (GC-14B; Shimadzu, Kyoto, Japan). The gas flow rate at the gas sampling point was recorded daily at 8:15. Total gaseous emissions are expressed as follows.

$$E = \Sigma V \left( C_{sample} - C_{air} \right)$$

Where E is the cumulative emission (g), V is the ventilation rate (L/12 hours), $C_{sample}$ and $C_{air}$ are the concentrations (g/L) of samples and ambient air, respectively.

The temperatures inside and outside the measurement chamber were recorded every 30 min by a recording thermometer (Espec, Osaka, Japan).

## Chemical analysis of the manure and estimation of the Emission Factor (EF)

The total solids (TS), volatile solids (VS), and total Kjeldahl nitrogen (TKN) of the manure samples were analyzed according to standard methods [23]. For the measurements of

inorganic-N, pH, and electrical conductivity (EC), 5 g of the manure sample was put into a 50-mL tube with 40 mL of deionized water, shaken (200 rpm, 30 min) and centrifuged (3000 g, 20 min). The supernatant was collected and filtered (0.45-μm). The manure pH was measured using a portable electrode (AS One, Osaka, Japan) using the standard method with modifications [24]. The inorganic-N ($NH_4^+$, $NO_2^-$ and $NO_3^-$) concentrations in the supernatant were measured by colorimetrical method (Bio-Rad, Hercules, CA).

Emission factors (EFs) for $CH_4$ and $N_2O$ were calculated by using cumulative $CH_4$ or $N_2O$ emission determined by GC measurement and total VS or N in the treated manure. These EFs were expressed as g $CH_4$ kg $VS^{-1}$ and g $N_2O$-N $kg^{-1}$ $N_{initial}$.

## DNA extraction and microbial community analysis

Manure samples were taken daily for each chamber (2 chambers with 2 runs, 4 replicates in total) and used for the microbial community analysis. DNA was extracted from 0.2 g of manure sample with an Isofecal Fecal DNA Extraction Kit (Nippon Gene, Tokyo) and a bead-beating system. DNA concentrations were measured using a Nanodrop™ Lite spectrophotometer (Thermo Fisher Scientific, Waltham, MA) and stored at −20˚C until further analysis.

A partial fragment of the 16S rRNA gene (the V4 hypervariable region) was amplified by a two-step polymerase chain reaction (PCR). Primers 515F and 806R [25] with Illumina adapter overhang sequences were used for the first-round PCR with 20 cycles, and indexes were attached to the amplicon with eight additional cycles. Each 20-μl PCR mixture contained 0.2 μL TaKaRa ExTaq HS DNA polymerase (TaKaRa Bio, Shiga, Japan) with 2 μL of buffer (10× buffer), 1.6 μL of 2.5 mM dNTP mix, 1 μL of each forward and reverse primer (10 mM), and 1 μL of template DNA.

The first-round PCR conditions were as follows: 94˚C for 2 min; 20 cycles of 94˚C for 30 s, 50˚C for 30 s, and 72˚C for 30 s; and a final 72˚C for 5 min. The PCR products were purified using an Agencourt AMPure XP purification system (Beckman Coulter, Indianapolis, IN) and then used for the second-round PCR with the following conditions: 94˚C for 2 min; eight cycles of 94˚C for 30 s, 60˚C for 30 s, and 72˚C for 30 s; and a final 72˚C for 5 min. Tag-indexed PCR products were purified again, and their quality and quantity were checked by an Agilent 2100 Bioanalyzer (Agilent, Santa Clara, CA) and a Qubit 2.0 Fluorometer and dsDNA HS Assay Kit (Life Technologies, Carlsbad, CA), respectively. Qualified amplicons were pooled in equal amounts and sequenced with a 250-bp paired-end sequencing protocol (Illumina, San Diego, CA).

Raw sequence reads were processed by Qiime2-2019.7 [26]. Paired-end sequences were merged and quality-filtered by DADA2 [27], and the de-noised feature table and amplicon sequence variants (ASVs) were used for a taxonomic diversity analysis. Taxonomic classification was assigned using a naïve Bayes classifier trained on the Greengenes 13_8_99% database, and mitochondria or chloroplast sequences were removed [28]. PICRUSt was used for predicting the function of the manure microbiome [29]. The closed-reference operational taxonomic units (OTUs) were normalized by copy number, and a new matrix of predicted functional categories was created with the KEGG database. STAMP was used to analyze the PICRUSt output file [30].

## qPCR assays of functional genes required for methanogenesis, nitrification, and denitrification

Quantitative PCR (qPCR) assays were performed with iTaq Universal SYBR Green Supermix and CFX96 (Bio-Rad) with a 20-μl reaction mix that contained 20 ng of template DNA. The primer pairs for amplifying the bacterial 16S rRNA gene, bacterial and archaeal amoA gene,

and bacterial denitrification genes (*nirS*, *nirK* and *nosZ*), and the PCR conditions for each reaction are summarized in S1 Table.

We prepared an external standard curve by using serial dilutions of a known copy number of the plasmid pGEM-T Easy vector (Promega, Madison, WI) containing each gene. The insert gene for the 16S rRNA gene and *nirS* was *Paracoccus denitrificans* (NCIMB 16712), and that for the AOB-*amoA* gene was *Nitrosomonas europaea* (NBRC 14298). Plasmids containing the cloned *nirK* gene (AB441832) and *nosZ* gene (AB441841) from dairy manure compost [31] and the cloned *nirS* gene, *mcrA* gene, and AOA-*amoA* gene from an environmental sample were used for the standard curve for these genes.

## Statistical analyses

All data were analyzed with SAS9.4 software [32]. We performed an analysis of variance (ANOVA) using the general linear model procedure. T-test or Tukey's multiple range comparison tests were used to separate the means. Probability (p)-values $<0.05$ were considered significant.

## Results

### Survey responses

The 20 farmers in Vietnam's Bentre province completed the survey. All were family farms with $5.0 \pm 1.5$ people per farm. The average number of workers per farm was $2.4 \pm 0.8$, with the working time of $5.8 \pm 1.1$ hours per day. The total cattle per farm was $9.3 \pm 4.4$ head, with an average body weight of $400 \pm 57$ kg. Lai Sind or Brahman crossbred *(Bos indicus)* were the main breeds. The cattle were fed elephant grass, Para grass, rice straw, and some other materials (S2 Table), with a dry matter intake (DMI) of 10.6 kg head$^{-1}$ day$^{-1}$ and crude protein (CP) level of 9.1% (per DM). All 20 farmers treated the manure from their cattle with a sun-drying method (S2 Fig). The farmers spread the manure on the ground or concrete floor to around 5 cm depth to maximize the drying efficiency. Six farmers (30%) used a biogas digester with the dirty water used for barn washes. None of the farms used composting or other manure treatment systems.

### Drying process, gas emissions, and mass balance

The weight of the manure dropped significantly from $87.7 \pm 13.2$ kg to $16.6 \pm 3.5$ kg during the drying process. Moisture loss occurred mainly from day 1 to day 4 by evaporation, and the weight was stable from day 5 to day 7 in both runs (S3A Fig). In contrast, the TS (%) increased significantly from $22.0 \pm 0.8\%$ to $96.7 \pm 0.6\%$. During this drying process, the losses of TS, VS, and N were very limited (S3B Fig), and most of the losses were explained by the sampling performed for the analysis. Since we separated the urine from the manure as much as possible, we did not have any leachate from the manure. The manure pH was also stable during the drying process, but there was a small difference between Run 1 (pH 8.0–8.5) and Run 2 (pH 7.5–7.6).

For both runs, methane emissions were detected at the beginning of the drying period, from day 1 to day 3 (Fig 1A). The N$_2$O emission was always at a background level during the 7-day drying period (Fig 1B). The emissions and mass balance during the sun-drying process are summarized in Table 1. The total CH$_4$ emission was $4.55 \pm 0.72$ g, which accounted for only 0.03% of the initial VS. The calculated CH$_4$ emission factor was $0.295 \pm 0.078$ g kg$^{-1}$ VS. Most of the VS contained in the initial manure (86.5%) remained in the final product. Most of the remaining VS could be explained by the sampling for the chemical analysis (12%).

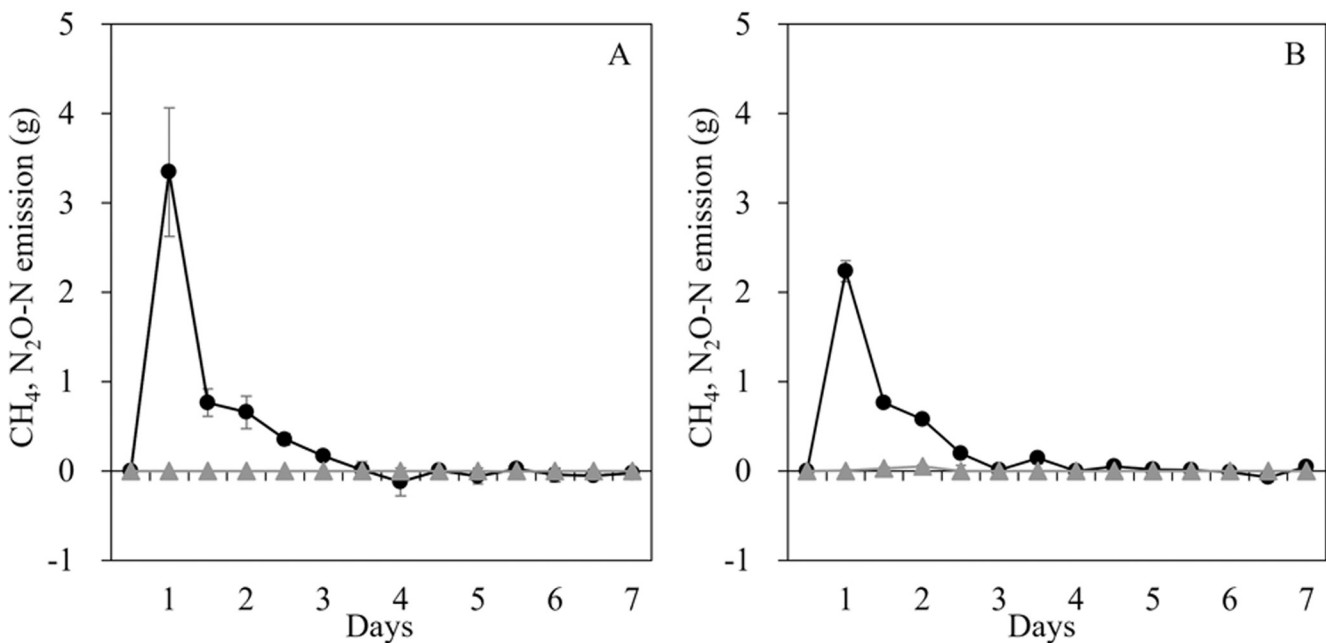

**Fig 1.** Methane (*circles*) and $N_2O$ (*triangles*) emission during the beef cattle manure sun-drying experiments in Runs 1 (**A**) and 2 (**B**). Error bars: SD (n = 2).

The total $N_2O$ emission was 0.046 ± 0.050 g, which accounted for 0.02% of the initial N, but it was always at a background level as stated above, indicating that the $N_2O$ emission was negligible during the beef cattle manure sun-drying process. Among the total N contained in the initial manure, 86.2% remained in the final dried manure, and 11.8% was explained by the sampling for the analysis.

The results of the inorganic-N measurements are illustrated in S4 Fig. Most of the inorganic N was in the $NH_4^+$-N state (1,321.4 ± 57.0 µg $g^{-1}$ TS in Run 1 and 821.3 ± 379.5 µg $g^{-1}$ TS in Run 2) at the beginning of the process in both runs. Only small amounts of $NO_2^-$-N (5.2 ± 4.0 µg $g^{-1}$ TS in Run 1 and 12.6 ± 3.3 µg $g^{-1}$ TS in Run 2) and $NO_3^-$-N (20.9 ± 19.0 µg $g^{-1}$ TS in Run 1 and 82.3 ± 98.0 µg $g^{-1}$ TS in Run 2) were detected on day 1. Most of the $NH_4^+$-N contained in the initial manure was lost over days 1–5 in both runs. In the last stage of the drying process, significant nitrification was detected in Run 2, and $NO_3^-$-N increased from 49.5 ± 63.0 to 528.1 ± 136.7 µg $g^{-1}$ TS over days 4 to 7 (S4B Fig). This was not observed in Run 1

**Table 1. Mass balance of the sun-drying of Vietnamese beef cattle manure (n = 4).**

| | VS (kg) | | % | N (kg) | | % |
|---|---|---|---|---|---|---|
| | Average | SD | | Average | SD | |
| Initial | 15.7 | 1.9 | 100.0 | 0.308 | 0.058 | 100.0 |
| Sampling | 1.9 | 0.1 | 12.0 | 0.036 | 0.003 | 11.8 |
| Final | 13.6 | 3.0 | 86.5 | 0.265 | 0.046 | 86.2 |
| $CH_4$, g | 4.55 | 0.72 | 0.03 | | | |
| $N_2O$, g | | | | 0.046 | 0.050 | 0.02 |
| Unknown | 0.2 | | 1.4 | 0.006 | | 2.0 |
| VS | | | | | | |

VS: volatile solids

where the increase in $NO_3^-$-N was only from $23.5 \pm 7.8$ to $55.3 \pm 40.0$ µg g$^{-1}$ TS during the same time period.

## Changes in the microbial community during the beef cattle manure sun-drying process

Bacterial and archaeal communities were monitored over time during the sun-drying process (Fig 2). A significant shift of the microbial community occurred at the beginning of the process, between days 0 and 2. At the phylum level, the relative abundance of *Proteobacteria* increased from $12.7 \pm 16.0\%$ to $34.6 \pm 4.3\%$ and that of *Actinobacteria* increased from $4.3 \pm 0.9\%$ to $10.2 \pm 1.1\%$, whereas the relative abundance of the following decreased: *Firmicutes*, from $49.9 \pm 14.6\%$ to $31.9 \pm 11.2\%$; *Bacteroidetes*, from $26.5 \pm 2.0\%$ to $18.8 \pm 5.0\%$; *Tenericutes*, from $2.6 \pm 1.3\%$ to $1.3 \pm 1.1\%$, and *Euryarchaeota*, from $1.2 \pm 0.1\%$ to $0.5 \pm 0.1\%$ (Fig 2A). During this period, the moisture content in Runs 1 and 2 fell from $78.3 \pm 1.3\%$ to $62.4 \pm 7.0\%$ and from $77.8 \pm 0.4\%$ to $50.9 \pm 14.0\%$, respectively (S3 Fig).

At the order level, the relative abundance of the following increased in both runs: *Xanthomonadales*, from $0.1 \pm 0.1\%$ to $2.1 \pm 1.0\%$; *Alteromonadales*, from $1.2 \pm 1.7\%$ to $6.8 \pm 2.8\%$; *Burkholderiales*, from $0.9 \pm 1.1\%$ to $3.6 \pm 1.8\%$; and *Actinomycetales*, from $2.5 \pm 3.0\%$ to $9.9 \pm 1.2\%$. The relative abundance of the following decreased in both runs: *Clostridiales*, from $41.7 \pm 20.3\%$ to $17.2 \pm 1.2\%$; *Bacteroidales*, from $22.4 \pm 7.75\%$ to $8.8 \pm 6.8\%$; and *Methanobacteriales*, from $1.1 \pm 0.1\%$ to $0.4 \pm 0.2\%$ (Fig 2B). However, for many orders an opposite trend was observed (e.g., *Bacillales* and *Pseudomonadales*), demonstrating that the effects of drying stress on the beef cattle manure microbial community are not consistent. The results of the principal component analysis (PCA) are illustrated in Fig 2C, showing the significant change from day 0 to day 2 in both runs. In addition, the microbial communities in Runs 1 and 2 fell into different regions, reflecting the inconsistent results for several orders.

The results of our analysis of the functional genes required for methanogenesis (*mcrA*) and the nitrification-denitrification process (*amoA*, *nirK*, *nirS*, *nosZ*) by qPCR assay are shown in Fig 3. We observed a temporal reduction of the total bacteria abundance in both runs in the middle stage of the process. The abundance of methanogens was higher in Run 2 ($4.5 \times 10^8$ to $1.1 \times 10^{10}$ copies g$^{-1}$ TS) compared to Run 1 (under the detection limit to $2.1 \times 10^8$ copies g$^{-1}$ TS), counter to $CH_4$ emission results. The abundance of both AOB-*amoA* and AOA-*amoA* was low in both runs, ranging from $1.1 \times 10^5$ to $9.3 \times 10^7$ copies g$^{-1}$ TS, which agrees well with the amplicon sequencing data, which did not detect any AOB sequences and only a very low level of AOA sequences in the whole community. Higher abundances of genes required for denitrification (*nirK*, *nirS* and *nosZ*) rather than nitrifiers were detected in both runs, up to $1.4 \times 10^{11}$ copies g$^{-1}$ TS.

## Discussion

### Emission of $CH_4$ and $N_2O$

According to the country's General Statistics Office, the majority (98.9%) of the farmers in Vietnam are small holders with 1–10 cattle, with a total of 5.8 million head in the entire country [33]. Most of the cattle are beef cattle, with dairy cattle (a mix of local breeds with Holstein) accounting for only 5.1%. We distributed a small-farm survey to investigate the typical beef production system in the southern region of the country, and the survey responses showed that all 20 of the participating farmers use sun-drying for their manure management. All farmers remove the manure from cattle barns every day, and they dry the manure in their backyard near the cattle barn. The rain occurs only 1–2 hours per day in the rainy season (May to

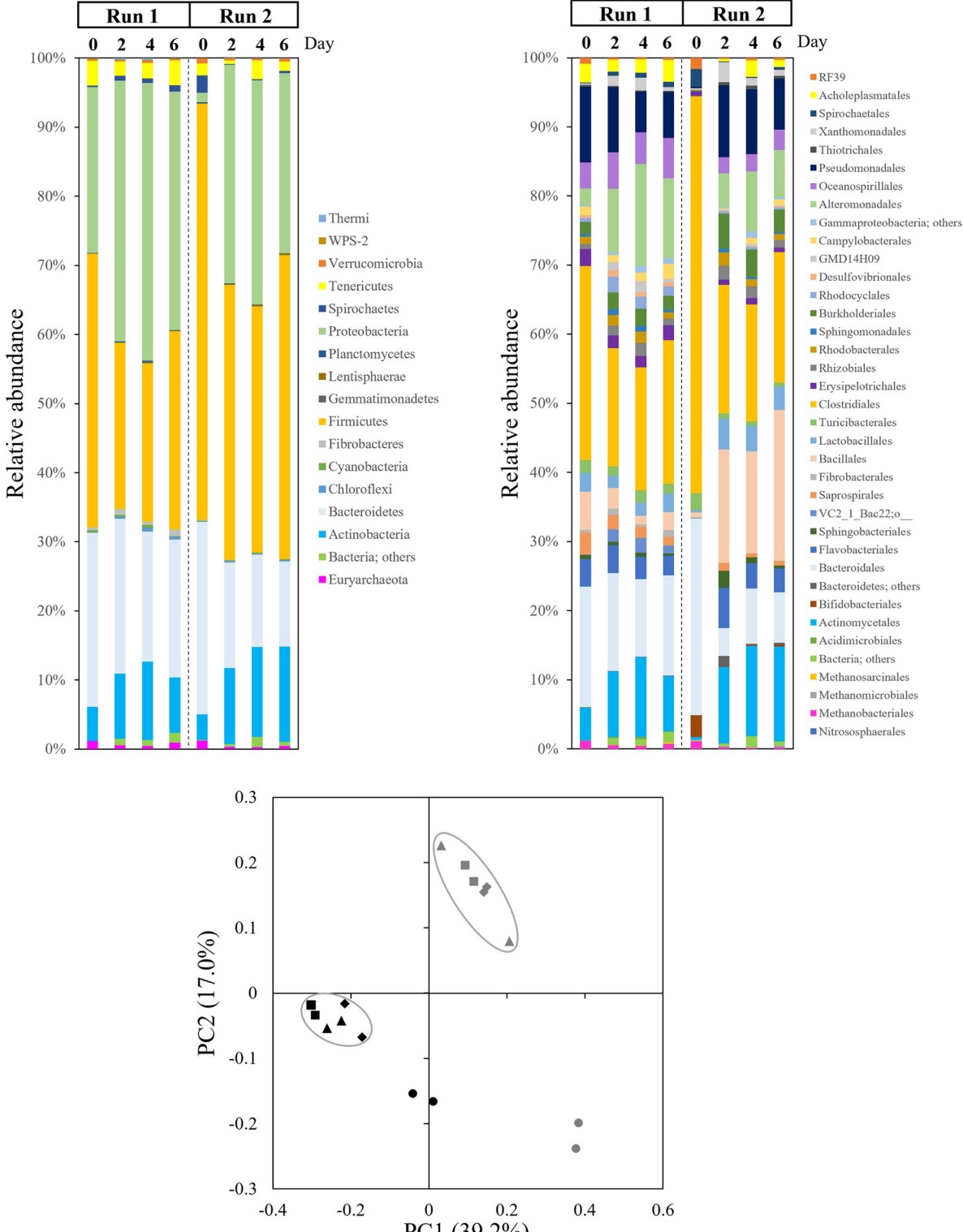

**Fig 2.** Changes in the bacterial/archaeal community at the phylum level (**A**) and order level (**B**), and the results of the principal component analysis (PCA) (**C**) during the sun-drying experiments. *Black symbols* = Run 1; *Gray symbols* = Run 2; *Circles* = week 1; *triangles* = week 3; *squares* = week 5; *diamonds* = week 7.

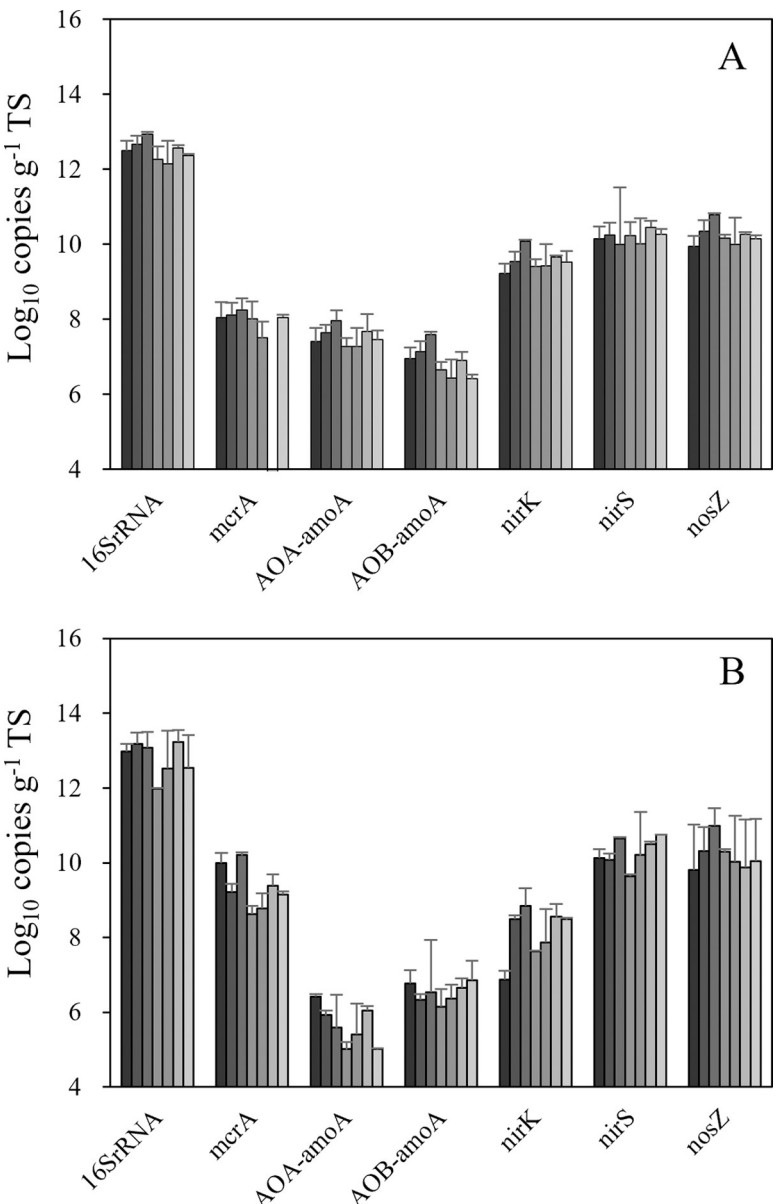

**Fig 3. Changes in the abundance of the marker gene (16SrRNA) and functional genes (*mcrA*, *amoA*, *nirK*, *nirS* and *nosZ*) required for CH$_4$ and N$_2$O emission during the sun-drying of the manure. A:** Run 1. **B:** Run 2. Error bars: SD (n = 4). The color gradient of the bars indicates the days 1 to 7 (from left to right).

October in southern Vietnam), therefore, the farmers can dry the manure under the sunlight the whole year. In the typical case, the farmers dry the manure for 2–3 days, and they rake the manure into the pile and cover it with the plastic seat when it rains. Since most of the farmers do not have the facility to store the dried manure, they sell it to the middleman who brings the dried manure into other regions for sale. Most of the farmers do not use the dried manure as fertilizer, and most of the dried manure produced in the region will be transported and used for perennial crops such as coffee, pepper, or dragon fruit. These results revealed that the sun-drying method is the dominant manure management system, at least in Southern Vietnam. However, since our farm survey only focused on a single province, more comprehensive surveys which cover the whole country will be needed in future studies.

In the IPCC guideline, sun-drying is described as a "Dry lot" system, and its default emission factor for $CH_4$ ranges from 1.0% to 2.0% [20] depending on the temperature, since the temperature greatly affects methanogenesis and higher temperatures can lead to higher emissions [34, 35]. These default values were derived from a study of dairy manure samples (5 kg) in small polyethylene containers with lids; these conditions are quite different from the sun-drying process used by Vietnamese beef cattle farmers [36]. More recently, the default value provided by IPCC was substantially updated with the observations from a comprehensive and systematic review on this topic [37, 38]. However, this recent update only covers the solid storage of manure in different situations (i.e., using a bulking agent or other additives, covered or compacted) and the categories "Dry lot" and "Daily spread" are still based on the old data.

In our present investigation, we measured the GHG emission directly with a chamber-based approach and provided the emission factor, which enables Tier 2 estimation for the category of sun-drying. Fresh beef manure from cattle fed a typical diet was spread on plastic sheets to mimic the farmers' methods (S2 Fig), and we detected $CH_4$ emission only at the beginning of the drying period with high reproducibility (Fig 1). This $CH_4$ emission accounted for only 0.03% of the initial VS (Table 1), and the estimated emission factor was $0.295 \pm 0.078$ g $kg^{-1}$ VS, which is considerably lower than the IPCC default value, indicating that the current approach provides an overestimation, at least for Mekong delta region in southern Vietnam, which experiences an average temperature of 24.5°C throughout the year. Since the technique of the different farmers was very similar—i.e., spreading the manure on the ground or concrete floor very thinly to a depth of around 5 cm—the different amounts of manure will affect only the total surface area used for the sun-drying. In this regard, our values can be used for different amounts of manure by adjusting the surface area used for spreading.

In a previous study of dairy cattle manure, which is still used for the current IPCC default value, the TS of the manure was much lower (14%) than that observed herein ($21.7 \pm 1.3\%$ in Run 1 and $22.2 \pm 0.4\%$ in Run 2), and the moisture content is much higher in dairy cattle manure [36]. Moreover, in the previous study, water was further added to increase the moisture and make a slurry, with a final TS value of 9%. Those experimental conditions are quite different from the actual sun-drying in Southeast Asian countries under different climate conditions, since our data establish that a significant decrease in the moisture content occurs at the first 2–3 days of sun-drying (S3 Fig). The methanogens are active under extreme reducing conditions, whereas a manure drying system can make the manure dry and introduce fresh air into the manure inside the system. These differences in conditions may explain the lower $CH_4$ emission values in the present study compared to the literature.

Our results also demonstrated that $CH_4$ emission occurs only at the beginning of the drying process. This strongly indicates that the sun-drying and moisture loss of the manure can reduce the emission of GHG from beef cattle manure. Although we could not measure the GHG emission during the transport of the manure from the pen to the chamber, our data clearly indicate that shortening the manure storage period in the pen or farmyard and immediate sun-drying can significantly minimize the emission of $CH_4$ from beef cattle manure.

Although there were no rainy days during the experimental period, the differences between Vietnam's two seasons (dry and rainy) must be considered. In the rainy season, the increase of humidity may delay the drying of the manure, which may prolong the $CH_4$ emission. The expected difference in ambient humidity might be small, but it would be better to measure the GHG emission in both seasons in further studies.

Moreover, we did not detect any significant $N_2O$ emission from the sun-drying process in either of the runs, with an estimated emission factor of $0.132 \pm 0.136$ g $N_2O$-N $kg^{-1}$ $N_{initial}$. Since we separated the urine as much as possible, N content in the manure used for the experiment was low. This could be the reason why we did not have significant $N_2O$ emissions during

the drying period. The emission factor value calculated in this study is far lower than the current default emission factor (2%), which puts sun-drying in the third highest $N_2O$ emission category among all manure management types, following composting with intensive windrows (10%) and the active mixing of a deep bedding system (7%) [20]. The separation of the urine and low N content in the manure also explains that the loss as $NH_3$ seems to be very limited since unknown N loss including $NH_3$ evaporation accounts for only 2% (Table 1). The estimated emission factor value came from the measurement of dairy (Brown Swiss) manure storage [39], which was performed under different circumstances from the actual sun-drying in Southeast Asian countries with different climate conditions. A similar example from a Japanese poultry manure drying facility shows that only $0.33 \pm 0.30\%$ of the total initial N was emitted as $N_2O$, which was detected only in the summer [40]. This value is still much higher than our present finding, indicating that the current default emission factor provided by the IPCC can lead to a significant overestimation of the $N_2O$ emission from the Asian manure management systems. The estimation of GHG emissions from Vietnamese livestock manure management (8.12 Tg $CO_2$eq.$yr^{-1}$ in 2014) [41] should therefore be revised based on our present estimation or future research which covers the manure management system in the whole country.

## Manure microbial community and functional gene abundance related to GHG emission

Since $CH_4$ and $N_2O$ are produced by the microbes present in the manure, both quantitative and qualitative microbiological data can help increase our understanding of their emissions. In this study, we monitored the composition of the total bacterial/archaeal community by 16SrRNA gene amplicon sequencing (Fig 2), and we observed that a significant shift occurred in the initial period (days 0–2). It seems likely that this change is attributable to the significant loss of moisture. The effect of the changes in pH or EC seems likely limited since no significant changes in these parameters were observed in this initial period.

The total microbial biomass measured by 16SrRNA gene quantification using real-time PCR also showed a temporal decrease over days 3 to 4 in both runs (Fig 3), which is also an effect of severe drying stress on these microbes. There have been extensive investigations of the effects of drying stress on environmental microbial communities, but most of them focused on the root microbiome, which is associated with the host plant and its physiology [42]. Research concerning the response of the bulk soil (apart from the plant root) microbiome to drying stress showed an increased relative abundance of *Actinobacteria* or other phyla [43] and decreased relative abundance of *Bacteroidetes* or many other groups of the phylum *Proteobacteria* [44], which agrees in part with our present observations. Considering these past and present results together, it can be said that severe drying stress by sun-drying reshaped the cattle manure microbial community in the present experiment.

Another effect of sun-drying reducing manure moisture content is the increase in the oxygen concentration inside the manure particles. Methanogens are strict anaerobes and are very sensitive to oxygen exposure [45, 46]. Therefore, a higher oxygen concentration seems to have the striking effect of reducing the methanogenesis activity, which supports our observation that the emission of $CH_4$ occurred only at the beginning of the drying process. The composition of methanogens was not consistent in the two runs in this study, indicating that not the methanogens composition but the relative abundance affects the $CH_4$ emission from the manure. The majority of the methanogens were of the genus *Methanobrevibacter*, which is also the dominant group in cattle rumen fluid [47]. A methylotrophic methanogen which is a close relative of *Methanomicrococcus blatticola* was detected only in Run 1 [48], whereas all of the methanogens in Run 2 were hydrogenotrophic methanogens (S5 Fig).

The estimated function of the microbial community also suggests that the methane metabolism decreased significantly between days 0 and 2 (S6 Fig). Although it was only in Run 1 and a very small portion of the total sequence (<0.1%), it is also noteworthy that a small amount of a potential methylotroph (order *Methylophilales*) was detected during the latter half of the drying process [49], indicating that slight methanotrophic activity also existed in the process. Although the methanogen could still be detected after the 7 days of drying by the amplicon sequencing data (which did not agree well with the qPCR assay results for the *mcrA* gene [Fig 3]), all of these data indicate that $CH_4$ emission occurred only at the beginning of the drying period, and methanogens were completely inactivated in the latter half of the process due to the loss of moisture and the higher oxygen concentration in the manure.

Regarding the emission of $N_2O$, the abundance of nitrifiers for both ammonia-oxidizing bacteria (AOB) and ammonia-oxidizing archaea (AOA) was low throughout the drying process, which agrees well with the qPCR results targeting the *amoA* gene for both groups (Fig 3). The results of the amplicon sequence also show that no AOB were detected throughout both experiments, and only 2 of a total 1.46 million sequences were detected as AOA (*Nitrososphaera*). The results of the $NO_2^-$ and $NO_3^-$ measurements revealed that only small amounts of these ions were detected in Run 1, whereas a significant amount of $NO_3^-$ was detected in the latter half of Run 2 (S4 Fig), which does not agree well with the qPCR data for the *amoA* gene.

Significant numbers of denitrifiers exist in raw manure, and these amounts did not decline through the sun-drying. Nitrate was observed in Run 2, but the $N_2O$ emission was under the detection limit. Our estimation of the function of the microbial community showed that the N metabolism tended to be increased between days 0 and 2, but the increase was not significant (S6 Fig). These results may explain why (1) we did not detect any significant $N_2O$ emission throughout the measurement period, and (2) only some nitrification occurred under the oxic conditions, especially in the latter half of the process.

## Application of dried manure to coffee or dragon fruit trees

The responses to our farm survey revealed that most of the farmers put their dried cattle manure in plastic bags and sold it to middlemen. Our interview of some of these middlemen showed that there is a flow of the dried manure into the central highlands (five provinces from Kon Tum to Lam Dong) or coastal region (seven provinces from Quang Nam to Binh Thuan). Other investigators identified similar manure flows from the coastal region (Binh Dinh and Phu Yen province) to the central highlands (Gia Lai, Dak Lak, and Dak Nong) or dragon fruit farmers in another coastal area (Binh Thuan) [19].

The second-largest region for coffee production worldwide is located in Vietnam's central highlands with poor soil quality, which require significant amounts of organic fertilizer to improve the soil physical properties. Four provinces in the highlands (Dak Lak, Lam Dong, Dak Nog, and Gia Lai) produce 92.4% of the nation's robusta coffee. Currently, 80% of the coffee farmers use dried beef cattle manure as organic fertilizer, and 69.3% of the farmers use coffee husks mixed with the manure (personal communication). A similar situation is seen for the dragon fruit farmers in the coastal area. There are also beef cattle in these central highlands (771,100) and coastal region (1,278,000). The numbers of cattle in these regions are higher than those in the southeast (six provinces from Binh Phuoc to Ho Chi Minh city; 394,900) and the Mekong delta (13 provinces from Long An to Ca Mau; 748,400). However, the supply of dried cattle manure in these regions seems insufficient to meet the farmers' demand.

The price of dried manure jumps from 5,000 Viet Nam Đồng (VND)/bag to ≥21,000–29,000 VND/bag due to the cost of transport by truck on trips taking up to 13 hours, but the

coffee farmers still purchase the manure because of their significant need. The entire scenario of this flow of dried manure is still not clear, since the Vietnamese government does not have the official statistical data. Further studies are needed to elucidate the whole structure. Another important issue is that the coffee farmers' application of dried manure to coffee trees could be an alternative source of GHG, for the following reasons.

First, our present findings demonstrate that organic matter in the manure is not lost during the drying period. Second, the methanogens seem to be inactivated due to the oxic conditions due to moisture loss, but we were able to detect methanogens after the 7-day drying period. Third, N contained in the manure also remains in the final dried manure, and we obtained evidence that nitrification could occur during the drying process. This could be a potential source of $N_2O$ after the application of the manure to coffee trees, as it would be embedded in the soil and can create anoxic conditions (which is favorable for the denitrifiers, the population of which did not decline throughout the entire drying process).

In conclusion, our farm survey demonstrated that sun-drying is the dominant beef cattle manure management system in Vietnam. In this process, the estimated emission factors for $CH_4$ and $N_2O$ were $0.295 \pm 0.078$ g kg$^{-1}$ VS and $0.132 \pm 0.136$ g $N_2O$-N kg$^{-1}$ $N_{initial}$, respectively, which are lower than the current default values provided by the IPCC for a Tier 1 approach. The sun-drying process induced a significant shift of the total microbial community, which may be attributed to a lower moisture content with significant drying stress. The relative abundance of the hydrogenotrophic methanogen *Methanobrevibacter* also fell significantly at the initial stage of the drying process, supporting that the emission of $CH_4$ occurred only at the beginning of the drying process. Although the abundance of detected nitrifiers' genes was very low, both $NO_2^-$ and $NO_3^-$ were detected. However, these nitrification activities did not lead to significant $N_2O$ emission.

## Supporting information

**S1 Fig. Chamber system for the gaseous emission measurement from manure sun-drying.**
(TIF)

**S2 Fig. Manure sun-drying by beef cattle farmers in Bentre province, Vietnam.**
(TIF)

**S3 Fig. Changes in the physicochemical parameters during the Vietnamese beef cattle manure sun-drying experiment. A:** Weight, *solid lines*; total solids (TS), *dashed lines*. **B:** Total solids (TS), solid lines; volatile solids (VS), *dashed lines*; total nitrogen (N), *dotted lines*. **C:** pH, *solid lines*; electrical conductivity (EC), *dashed lines*. *Black squares*: Run 1. *Gray circles*: Run 2. Error bars: standard deviation (SD) (n = 2).
(TIF)

**S4 Fig. Changes of the inorganic-N ($NH4^+$: Dark gray bars; $NO_2^-$: Light gray bars; $NO_3^-$: Black bars) during the manure sun-drying experiments.** Error bars: SD (n = 2). TS: total solids.
(TIF)

**S5 Fig. Relative abundance of methanogens during the beef cattle manure drying process.**
A and B indicate runs 1 and 2, respectively. Light gray bars: *Methanobacteriales*. Dark gray bars: *Methanosarcinales*. White bars: *Methanomicrobiales*.
(TIF)

**S6 Fig. Changes in the functional abundance relative to methane and nitrogen metabolism during the beef cattle manure sun-drying process estimated by PICRUSt.**
(TIF)

**S1 Table. Primer sequence and PCR conditions for 16S rRNA, AOB-*amoA*, AOA-*amoA*, *mcrA*, *nirK*, *nirS* and *nosZ*.**
(DOCX)

**S2 Table. Chemical composition of feeds of the beef cattle in Bentre province.**
(DOCX)

**S3 Table. Bacterial diversity during sun-drying of Vietnamese beef cattle manure.**
(DOCX)

**S4 Table. Change of the manure chemical property during sun-drying experiment.**
(DOCX)

## Acknowledgments

The authors are grateful to Dr. Tobita for his contribution to the farm survey and useful discussions. This work was implemented under the Japan International Research Center for Agricultural Sciences project 'Climate Change Measures in Agricultural Systems'.

## Author Contributions

**Conceptualization:** Koki Maeda.

**Data curation:** Koki Maeda.

**Formal analysis:** Koki Maeda.

**Investigation:** Van Thanh Nguyen, Koki Maeda, Yukiko Nishimura, Trinh Thi Hong Nguyen, Dien Duc Nguyen, Tomoyuki Suzuki.

**Methodology:** Koki Maeda.

**Project administration:** Trinh Thi Hong Nguyen, Kinh Van La.

**Resources:** Dien Duc Nguyen.

**Supervision:** Kinh Van La.

**Writing – original draft:** Van Thanh Nguyen, Koki Maeda.

**Writing – review & editing:** Koki Maeda.

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
