## [Decision Letter · Decision Letter 0]

4 Nov 2021

PONE-D-21-31577Emission factors for Vietnamese beef cattle manure sun-drying and the effects of drought stress on manure microbial communityPLOS ONE

Dear Dr. Maeda,

Thank you for submitting your manuscript to PLOS ONE. After careful consideration, we feel that it has merit but does not fully meet PLOS ONE’s publication criteria as it currently stands. Therefore, we invite you to submit a revised version of the manuscript that addresses the points raised during the review process.

We look forward to receiving your revised manuscript.

Kind regards,

James E. Wells, PhD

Academic Editor

PLOS ONE

Journal Requirements:

3. We note that Figure S2 in your submission contain copyrighted images. All PLOS content is published under the Creative Commons Attribution License (CC BY 4.0), which means that the manuscript, images, and Supporting Information files will be freely available online, and any third party is permitted to access, download, copy, distribute, and use these materials in any way, even commercially, with proper attribution. For more information, see our copyright guidelines: http://journals.plos.org/plosone/s/licenses-and-copyright.

a. You may seek permission from the original copyright holder of Figure S2 to publish the content specifically under the CC BY 4.0 license. 

“I request permission for the open-access journal PLOS ONE to publish Figure S2 under the Creative Commons Attribution License (CCAL) CC BY 4.0 (http://creativecommons.org/licenses/by/4.0/). Please be aware that this license allows unrestricted use and distribution, even commercially, by third parties. Please reply and provide explicit written permission to publish Figure S2 under a CC BY license and complete the attached form.”

Reviewers' comments:

Reviewer's Responses to Questions

**Comments to the Author**

1. Is the manuscript technically sound, and do the data support the conclusions?

Reviewer #1: Partly

Reviewer #2: Yes

2. Has the statistical analysis been performed appropriately and rigorously? 

Reviewer #1: I Don't Know

Reviewer #2: Yes

3. Have the authors made all data underlying the findings in their manuscript fully available?

Reviewer #1: Yes

Reviewer #2: No

4. Is the manuscript presented in an intelligible fashion and written in standard English?

Reviewer #1: Yes

Reviewer #2: Yes

5. Review Comments to the Author

Reviewer #1: General: It would seem the current emission factors estimates are grossly overestimating GHG emission from beef farms in Vietnam, thereby supporting the need for this research. My primary concern is very small number of experimental units for this study. The survey reached only 20 producers out of an estimated 580,000 (5.8 million cattle with average herd size of 10 cattle). And the emission studies were conducted on two runs with two replicates each (N=4). I would suggest additional replicates under different environmental conditions (rainy vs. dry season, hot vs colder, etc) to fully be able to conclude that the current emission factors need to be replaced.

Specific comments are included on attachment.

Reviewer #2: This is a very well written manuscript describing two manure drying studies focusing primarily on CH4 and N2O emissions and the changes in microbial populations during manure drying (predominant method in a Southern Vietnam province). Methods and conclusions are appropriate. My primary concerns are:

1. the limited degree of replication, and

2. the limited scope of the emission data (in the context of the larger manure system).

I feel that both of these can easily be addressed by including a little extra text in the manuscript. This is a great initial investigation worthy of publication.

Limited degree of replication--only two runs were conducted with multiple sampling of each experimental unit. From my read of the manuscript (and figure/table legends), it appears that gas emissions were measured twice on each tarp on sampling dates and that 4 samples were collected on those dates. This seems to be a split sample, each repeatedly measured for gases, and each sample was sampled at two locations (2 x 2 = 4 total manure samples). Take a few sentences in the experimental description to clarify how you set up the experiments and the number of sites sampled for both gases and manure. Please include the length of the drying and that daily sampling was done for 1 week. Also in the microbiology section, what does the 'manure sample (line 155) represent? Is it a composite from all four locations?

Emission data scope--really this is some initial data into gas emissions from drying manure over 1 week. Really how representative is this by the farmers? Do they rake daily and always cover with a tarp? What happens if it gets rained on? What happens if they dry it for two weeks? This is a very narrow set of conditions investigated. You should really point this out in the final discussion paragraph and build a case for repeating this study with a range of potential conditions that farmers may encounter and management choices. Also, is the fresh manure removed daily? I think CH4 degassing from fresh manure (even over a few hours) would be substantial. This is good initial data that should be supplemented with additional studies. Make a good case for this in the paper.

Specific comments:

Title--'drought stress' by microbes is not a very good term. There is a rich body of work looking at drought stress, which is a prolonged soil drying and its effects on plants. I think 'effects of manure drying on the microbial community' is a better phrase for the title.

L 148 'measured by colorimetric method'

L 107 Is urine separation by sloping concrete floor typical of farmers in Southern Vietnam? This may explain your low N2O emissions if you remove easily oxidized N from the manure stream.

L 188 'Quantitative PCR'

L 214 A very high forage diet! Is this typical for cattle in S. Vietnam? Does the diet change as resources change? Might be good to explore diet effects on emissions (or talk about that for future work).

L228 'losses explained by sampling? I don't think that is likely. You are missing both CO2 emissions and potentially leachate losses. Also NH3 loss can be substantial (but is lower for high forage diets). Maybe just characterize 'Sampling Losses' as 'Other' and then mention CO2 emissions and NH3 emissions and potential leachate losses (or other sources).

L 250 - same comment at L228

L 301 'TS), counter to CH4 emission results.'

L 325 YES! bring out that this is initial data for a single province. Need to build on this study to study manure handling in other areas.

L 346 Reign in how broadly you apply your results--'at least for a province of Southern Vietnam.'

L 390 to 392 I don't think your two manure drying runs is enough evidence to revise an IPCC factor. Your data is great justification for further work to determine if the factor is an overestimation. You also don't look at the whole manure management component. When applied to soils for fertilizer, you could have a very large gas efflux as new C is degraded in the cropping systems.

L 404 'effect of drying stress on' Avoid the use of 'drought' and use 'drying' instead

6. PLOS authors have the option to publish the peer review history of their article (what does this mean?). If published, this will include your full peer review and any attached files.

Reviewer #1: No

Reviewer #2: No

---

## [Author Response · Author response to Decision Letter 0]

12 Jan 2022

General: It would seem the current emission factors estimates are grossly overestimating GHG emission from beef farms in Vietnam, thereby supporting the need for this research. My primary concern is very small number of experimental units for this study. The survey reached only 20 producers out of an estimated 580,000 (5.8 million cattle with average herd size of 10 cattle). And the emission studies were conducted on two runs with two replicates each (N=4). I would suggest additional replicates under different environmental conditions (rainy vs. dry season, hot vs colder, etc) to fully be able to conclude that the current emission factors need to be replaced. 

"Thank you very much for your helpful comments. We selected 20 farmers which have to be the representative activity in the Ben Tre Province, where the highest beef cattle farmers number (176.2 thousand heads) in the region. We agree with the reviewer that a more extensive farm survey will be required for the more robust dataset, but we are confident that the manure management practice identified with this survey is very much widespread in the region.

We agree with the reviewer that it would be better to provide more data for the rainy season, as we already stated in the original manuscript (L370). However, generally, most of the farmers do not sun-dry the manure even in the rainy season, therefore, we believe that our dataset covers the major part of the manure management system in the region."

Line 43: Are you saying Vietnam ranks third in beef production among the Association of South-East Asian Nations? Yes. We obtained the data at FAOSTAT (2019), https://www.fao.org/faostat/en/#data/QCL

Line 93: Did you have 100% participation from those asked to complete the survey? How did you choose the 20 farms you selected? Do you feel this is representative of approximately 580,000 farms? "All farmers completed the questionnaire. The region chosen to make the survey was the highest numbers density of cattle in Southern Vietnam. Some previous studies also stated that drying cattle manure is the most common method in Vietnam, which is already stated in the original manuscript (L72).

Reference: Dan et al. Project report, 24-27 (2004), McRoberts et al., Agr. Food. Sys., 33;86-101 (2017)"

Line 104 and 105: Suggest changing “farmers’ behavior” to “producers’ practices”. Thank you very much for your suggestion. We changed the words here at L104 and 106.

Line 106: Were there two replicates for each run for a total N =4? That’s not many experimental units. "Yes, we had two runs with two replicates (N=4). According to both reviewers' comments, we added some sentences to describe the experimental setup and sampling procedures clearly. (L114-121)

""We had two replicates (chambers) for each run, and the same experiments were done twice (Runs 1 and 2). During the 7 days experimental period, the manure was dried in the sun all day and covered all night with tarpaulins. No rain fell during the experimental period. Manure samples were collected at 12:50 p.m. every day at five points (four at corners and one at center) in each chamber, mixed well, and kept in a freezer at −20°C for until analysis. The weights of the samples were recorded every time to enable the calculation of the loss of by sampling precisely."""

Line 108: Do you mean just the feces or the manure (mixture of feces and urine)? Urine was excluded as much as possible, and only feces (manure) was used for the experiment. We collected fresh manure immediately after cattle were released to ensure urine was not included in manure. (L108)

Line 151: Maybe I missed it, but how did you determine the volatile solids content? We analyzed the VS content in the manure by the standard method, which is stated in L141 in the original manuscript. Briefly, manure was dried under 105C for 24 h (for TS determination), and dried manure was further processed under 600C for 1 h. Weight of ash and dried manure was used for VS calculation.

Line 202: What were the variables in your statistical model? We performed the statistical analysis on manure physicochemical parameters and NGS data. Proc GLM was performed for the diversity of manure microbiome, and these results are presented in Table S3. We had classes run and day for this dataset. We had two replicates for each run, without any treatments. 

Line 214-216: Were all the cattle from all the farms fed the same diet? If not, how did you account for differences in nutrient composition of the manure from the cattle fed different diets? Yes, all cattle fed same diet was used, which is stated in the original manuscript at L108-110.

Line 373: You state the increase in humidity may reduce or delay the drying of manure, which may lead to delayed moisture loss and an inhabitation of CH4 emission. However, would higher moisture content increase anaerobic conditions in the manure which would increase CH4 production and emission? "Thank you very much for your pointing. We changed the sentence to avoid misleadings (L 390-392.) 

""In the rainy season, the increase of humidity may delay the drying of the manure, which may prolong the CH4 emission."""

Line 385: Suggest changing to read, “A similar example…” Thank you very much for your pointing. We revised the text as you suggested. (L407)

Line385-387: Beef should be even lower than poultry since poultry manure typically has very high N content compared to beef. "Thank you very much for your comment. We stated that urine was separated and only manure was used for the experiment, which is an alternative reason why we did not detect significant N2O emission. (L396-399)

""Since we separated the urine as much as possible, N content in the manure used for the experiment was low. This could be the reason why we did not have significant N2O emissions during the drying period. """

Line 419: Why do you think the composition of the methanogens was not consistent between the two run? How confident are you in your methane data with only N=4 and two runs? "Thank you very much for your comment.

Unfortunately, we cannot explain why we had the difference in methanogen composition between runs. Data presented in the S5 figure shows that the change in the relative abundance of methanogens was similar between runs. It dropped between days 0-2 were the same in both runs, and this change agrees with the CH4 emission data, which also dropped between days 0-2 (Fig. 1). Therefore, it can be said that not the composition of the methanogen does not affect the CH4 emission but the relative abundance affect it. We started this in the revised manuscript (L443-445).

Methane emission and methanogens abundance dropped in both runs, which is the consistent point in this study. We are confident that this is the general phenomenon when we dry the beef cattle manure under the sunlight. "

Line 400-401: You measured pH and EC but don’t discuss how these factors may have influenced GHG emissions. You seem to focus only on moisture and temperature. Thank you very much for this helpful comment. We did not see significant changes in pH or EC in the initial period (days 0-2) where we had significant reduction in CH4 emission. We added the sentence at line 424-426.

Line 453-489: This is an interesting discussion that seems loosely tied to this paper. It seem a bit out of place in this particular manuscript. Thank you very much for your suggestion. We found that most of the dried manure is transported into the central highlands or coastal area, where a significant amount of coffee, pepper, or dragon fruit is produced. Application of dried manure can be the alternative GHG source or it can cause other pollution, therefore, we believe that this paragraph should be accompanied by the main body to identify the future direction of our research.

"This is a very well written manuscript describing two manure drying studies focusing primarily on CH4 and N2O emissions and the changes in microbial populations during manure drying (predominant method in a Southern Vietnam province). Methods and conclusions are appropriate. My primary concerns are:

1. the limited degree of replication, and

2. the limited scope of the emission data (in the context of the larger manure system).

I feel that both of these can easily be addressed by including a little extra text in the manuscript. This is a great initial investigation worthy of publication." Thank you very much for your favorable and helpful comments. We revised our manuscript according to your comments and suggestions. We believe that our manuscript is now greatly improved based on your guidance.

Limited degree of replication--only two runs were conducted with multiple sampling of each experimental unit. From my read of the manuscript (and figure/table legends), it appears that gas emissions were measured twice on each tarp on sampling dates and that 4 samples were collected on those dates. This seems to be a split sample, each repeatedly measured for gases, and each sample was sampled at two locations (2 x 2 = 4 total manure samples). Take a few sentences in the experimental description to clarify how you set up the experiments and the number of sites sampled for both gases and manure. Please include the length of the drying and that daily sampling was done for 1 week. Also in the microbiology section, what does the 'manure sample (line 155) represent? Is it a composite from all four locations? "Thank you very much for your helpful suggestion.

We had two chambers for replicates, and the same experiments were done twice (2 chambers ×2 runs =4 gas measurements). We clearly stated this in the revised manuscript (L114). We collected the gas samples twice per day for 7 days, and ambient air samples were also taken at the same time for background measurement. Manure samples were also taken daily, after the mixing. It is already stated in the original manuscript. 

Manure samples in each chamber, taken dairy after mixing were used for the microbial community analysis. We started this in the revised manuscript to make it clear. 

""We had two replicates (chambers) for each run, and the same experiments were done twice (Runs 1 and 2). During the 7 days experimental period, the manure was dried in the sun all day and covered all night with tarpaulins. No rain fell during the experimental period. Manure samples were collected at 12:50 p.m. every day at five points (four at corners and one at center) in each chamber, mixed well, and kept in a freezer at −20°C for until analysis. The weights of the samples were recorded every time to enable the calculation of the loss of by sampling precisely."""

Emission data scope--really this is some initial data into gas emissions from drying manure over 1 week. Really how representative is this by the farmers? Do they rake daily and always cover with a tarp? What happens if it gets rained on? What happens if they dry it for two weeks? This is a very narrow set of conditions investigated. You should really point this out in the final discussion paragraph and build a case for repeating this study with a range of potential conditions that farmers may encounter and management choices. Also, is the fresh manure removed daily? I think CH4 degassing from fresh manure (even over a few hours) would be substantial. This is good initial data that should be supplemented with additional studies. Make a good case for this in the paper. "Thank you very much for this helpful comment. We added some sentences to describe the typical farmers' practice in the discussion section (L332-341). 

""All farmers remove the manure from cattle barns every day, and they dry the manure in their backyard near the cattle barn. The rain occurs only 1-2 hours per day in the rainy season (May to October in southern Vietnam), therefore, the farmers can dry the manure under the sunlight the whole year. In the typical case, the farmers dry the manure for 2-3 days, and they rake the manure into the pile and cover it with the plastic seat when it rains. Since most of the farmers do not have the facility to store the dried manure, they sell it to the middleman who brings the dried manure into other regions for sale. Most of the farmers do not use the dried manure as fertilizer, and most of the dried manure produced in the region will be transported and used for perennial crops such as coffee, pepper, or dragon fruit. """

Title--'drought stress' by microbes is not a very good term. There is a rich body of work looking at drought stress, which is a prolonged soil drying and its effects on plants. I think 'effects of manure drying on the microbial community' is a better phrase for the title. Thank you very much for your suggestion. We changed the title of our manuscript as you kindly suggested. "Emission factors for Vietnamese beef cattle manure sun-drying and the effects of drying on manure microbial community"

L 148 'measured by colorimetric method' Thank you very much for your pointing. We changed the sentence as you suggested. (L152)

L 107 Is urine separation by sloping concrete floor typical of farmers in Southern Vietnam? This may explain your low N2O emissions if you remove easily oxidized N from the manure stream. "Thank you very much for your helpful comment. The cattle barn with a sloping concrete floor is very typical in Vietnam. We agree with the reviewer that removing the urine can be one of the reasons why we did not detect significant N2O emissions. We added the sentence in the discussion section (L396-398).

""Since we separated the urine as much as possible, N content in the manure used for the experiment was low. This could be the reason why we did not have significant N2O emissions during the drying period. """

L 188 'Quantitative PCR' Thank you very much for your pointing. We corrected this error at L194.

L 214 A very high forage diet! Is this typical for cattle in S. Vietnam? Does the diet change as resources change? Might be good to explore diet effects on emissions (or talk about that for future work). This is the typical diet composition in southern Vietnam. We agree with the reviewer that diet change can affect GHG emissions. The purpose of this work is to provide the values which can be used for the emission factor value from this local manure management category. Therefore, we think that the values in this work can be a good candidate to be the representative value for a typical beef cattle manure management system.

L228 'losses explained by sampling? I don't think that is likely. You are missing both CO2 emissions and potentially leachate losses. Also NH3 loss can be substantial (but is lower for high forage diets). Maybe just characterize 'Sampling Losses' as 'Other' and then mention CO2 emissions and NH3 emissions and potential leachate losses (or other sources). "Thank you very much for your helpful comment.

We recorded the weight of the samples every time, and data TS or VS content was available for each sampling event. Therefore, we could estimate the loss by each sampling precisely. Data presented in Table 1 comes from those measurements, so the loss by sampling can be separated by ""unknown"". 

Although we did not measure CO2 emission, we think we did not have significant organic matter degradation during the process, because the loss of the moisture within the first 2-3 days significantly deactivated the activity of the microbes. In this condition, we should still have some CO2 emissions, and we think that those emissions should be included in the ""unknown"" loss described in the table since we did not have any measurements. Moreover, since we tried to separate the urine as much as possible, we did not have any leachate in these experiments. We added some sentences in the M&M section and discussion section (L119, L234). "

L 250 - same comment at L228 "Thank you very much for your helpful comment.

We added the sentence about the NH3 loss, which is included in the ""unknown"" N loss during the process to the discussion section. (L402)"

L 301 'TS), counter to CH4 emission results.' We changed the sentence as you suggested. (L311)

L 325 YES! bring out that this is initial data for a single province. Need to build on this study to study manure handling in other areas. Thank you very much for your comment. We think we need more comprehensive dataset which covers the whole country. 

L 346 Reign in how broadly you apply your results--'at least for a province of Southern Vietnam.' Thank you very much for your comment. We changed the sentence here as you suggested. (L364)

L 390 to 392 I don't think your two manure drying runs is enough evidence to revise an IPCC factor. Your data is great justification for further work to determine if the factor is an overestimation. You also don't look at the whole manure management component. When applied to soils for fertilizer, you could have a very large gas efflux as new C is degraded in the cropping systems. "Thank you very much for your helpful comment. We agree with the reviewer, but we believe that this dataset can be the initial description of the local manure management system. Further research will be required to make the value more robust. We added one sentence to state it in L414. 

For the comment on the manure application to the soil, we have already provided an additional paragraph in the original manuscript about the fate of the dried manure which is transported into the central highlands for coffee and pepper production. Application of the dried manure for these cropping systems can be the additional GHG emission source, and we will focus on that in future studies."

L 404 'effect of drying stress on' Avoid the use of 'drought' and use 'drying' instead Thank you very much for your comment. We replaced the word throughout the manuscript.

---

## [Decision Letter · Decision Letter 1]

7 Feb 2022

Emission factors for Vietnamese beef cattle manure sun-drying and the effects of drying on manure microbial community

PONE-D-21-31577R1

Dear Dr. Maeda,

We’re pleased to inform you that your manuscript has been judged scientifically suitable for publication and will be formally accepted for publication once it meets all outstanding technical requirements.

Kind regards,

James E. Wells, PhD

Academic Editor

PLOS ONE

Additional Editor Comments (optional):

Reviewers' comments:

Reviewer's Responses to Questions

**Comments to the Author**

1. If the authors have adequately addressed your comments raised in a previous round of review and you feel that this manuscript is now acceptable for publication, you may indicate that here to bypass the “Comments to the Author” section, enter your conflict of interest statement in the “Confidential to Editor” section, and submit your "Accept" recommendation.

Reviewer #1: All comments have been addressed

Reviewer #2: All comments have been addressed

2. Is the manuscript technically sound, and do the data support the conclusions?

Reviewer #1: Yes

Reviewer #2: Yes

3. Has the statistical analysis been performed appropriately and rigorously? 

Reviewer #1: Yes

Reviewer #2: Yes

4. Have the authors made all data underlying the findings in their manuscript fully available?

Reviewer #1: Yes

Reviewer #2: (No Response)

5. Is the manuscript presented in an intelligible fashion and written in standard English?

Reviewer #1: Yes

Reviewer #2: Yes

6. Review Comments to the Author

Reviewer #1: (No Response)

Reviewer #2: (No Response)

7. PLOS authors have the option to publish the peer review history of their article (what does this mean?). If published, this will include your full peer review and any attached files.

Reviewer #1: No

Reviewer #2: No

---

## [Editor Report · Acceptance letter]

8 Mar 2022

PONE-D-21-31577R1 

Emission factors for Vietnamese beef cattle manure sun-drying and the effects of drying on manure microbial community 

Dear Dr. Maeda:

I'm pleased to inform you that your manuscript has been deemed suitable for publication in PLOS ONE. Congratulations! Your manuscript is now with our production department. 

Kind regards, 

on behalf of

Dr. James E. Wells 

Academic Editor

PLOS ONE